# Survey on Fungi in Antarctica and High Arctic Regions, and Their Impact on Climate Change

Masaharu Tsuji [1,2]

1 Department of Materials Chemistry, National Institute of Technology (KOSEN), Asahikawa College, Asahikawa 071-8142, Japan; spindletuber@gmail.com
2 National Institute of Polar Research (NIPR), Tachikawa 190-8158, Japan

**Abstract:** The Antarctica and High Arctic regions are extreme environments, with average maximum temperatures below 0 °C for most days of the year. Interestingly, fungi inhabit these regions. This review describes the history of fungal surveys near the Syowa Station and the fungal diversity in this region. In the High Arctic region, I summarize the changes in the fungal communities of the glacial retreat areas of Ny-Ålesund, Norway and Ellesmere Island, Canada, in response to climate change. In addition, the ability of Antarctic and Arctic fungi to secrete enzymes at sub-zero temperatures is presented. Finally, the future directions of Antarctic and Arctic fungal research are provided.

**Keywords:** antarctica; high arctic; fungi; climate change

## 1. Introduction

The Arctic region is primarily covered by the Arctic Ocean, which is surrounded by land. In contrast, the Antarctic region is characterized by the Antarctic Continent, covered with ice and snow. Collectively, these two areas are referred to as the Polar Regions.

Antarctica is one of the planet's most extreme environments. It is exposed to cold and dry conditions, with the lowest temperature being –90 °C. Approximately 98% of its surface is covered with snow and ice, and the remaining 2% represents the ice-free area located in coastal areas and high mountains, wherein the snow melts in summer, exposing the ground. Most of the terrestrial ecosystems of Antarctica are distributed in this ice-free area [1]. Syowa Station, the headquarters for the Japanese Antarctic Research Expedition (JARE), is also an ice-free area.

The High Arctic, like Antarctica, is one of the planet's most extreme environments, with a maximum temperature of below 0 °C for most of the year.

Despite its exposure to adverse conditions, such as subzero temperatures and low nutrient and water availability, the fungi that inhabit cold environments can grow at near-subzero temperatures. The secretion of extracellular enzymes allows them to utilize complex materials as energy sources. Therefore, psychrophilic and psychrotolerant fungi play essential roles in the nutrient cycle of the polar region ecosystems [2].

This review presents the history of fungal surveys near the Syowa Station and the fungal diversity in this region. Furthermore, the impact of climate change on fungal diversity is summarized based on fungal surveys conducted in glacier retreat areas in the High Arctic. The growth and enzyme activity of the fungi inhabiting the polar regions at subzero temperatures are also introduced. Finally, the prospects of fungal research in the Arctic and Antarctic regions are discussed.

## 2. History of Fungal Research in the Antarctic Region and Near Syowa Station

Antarctica was first sighted in 1820, and the first landing occurred a year later [3]. The first report on fungi from this region was published circa 1897–1899 by a Belgian expedition that collected *Sclerotium antarcticum* from Danco Island near the Antarctic Peninsula [4]. The

late Roald Amundsen (1872–1928), who was the first to reach the South Pole, participated in this expedition.

In the 1960s, several new fungal species were reported by Fell et al. [5] and di Menna [6,7]. Bridge and Spooner (2012) created a "list of non-lichenized fungi from the Antarctic region", an essential tool to understand Antarctica's fungal diversity. However, the list still needs to be updated [8].

The first report on fungi around Syowa Station, the headquarters of the Japanese Antarctic Research Expedition (JARE), was published in 1961 [8], followed by three articles that reported twelve ascomycetes and four basidiomycetes (Table 1) [9–12].

**Table 1.** The number of fungal species near Syowa Station through 2012 and 2013–2022.

|  | **Through 2012** | **2013–2022** | **Total** |
| --- | --- | --- | --- |
| Chytridiomycota | 0 | 0 | 0 |
| Zygomycota | 0 | 0 | 0 |
| Ascomycota | 12 | 49 | 61 |
| Basidiomycota | 4 | 12 | 16 |
| Total | 16 | 61 | 77 |

In the 1960s, the JARE was suspended. Nevertheless, during this period, *Moesziomyces antarcticus* was isolated from Ross Island [13]. This fungal species secretes stable lipase and produces biosurfactants [14,15].

From the 1960s to 2013, no additional fungal species were reported near the Syowa Station by JARE. After this date, the activities of JARE revealed the partial diversities of fungi near the Syowa Station; this was performed by mycologists Dr. Tamotsu Hoshino and Dr. Takashi Osono, who participated in the 48th (2006–2007 austral summer) and 51st JARE (2009–2010 austral summer), respectively [16,17].

### 3. Fungal Diversity Research near Syowa Station

The samples collected during JARE–48 and JARE–49 allowed the survey of the fungal diversity in the Skarvsnes ice-free area and East Ongul Island (Figure 1). Each species name was checked. The species' names were updated using Index Fungorum (http://www.indexfungorum.org) (accessed on 19 September 2023). It should be noted that some isolates or strains were not identified at the species level in the original article. An attempt was made to reclassify these organisms if their DNA sequences had been deposited in a DNA databank. In cases in which reclassification at the species level was unattainable, the isolates or strains were presented at the genus level.

In the Skarvsnes ice-free area, 97 fungal strains were isolated and classified into 5 genera and 5 species of Ascomycota, as well as into 8 genera and 10 species of Basidiomycota. *Mrakia* spp. was the dominant genus (53%), corresponding to the species *M. blollopis* (23%), *M. gelida* (21%), and *M. robertii* (9%) (Figure 2) [16]. The genus *Mrakia* is a basidiomycete yeast previously isolated from various cold environments worldwide, including the Arctic region, the Himalayas, the Alps, and Antarctica [18].

On East Ongul Island, 196 fungal strains were isolated and classified into 8 genera and 9 species of Ascomycota, as well as 10 genera and 16 species of Basidiomycota. The dominant fungal species on the island were *Vishniacozyma victoriae* (23.2%), *Naganishia friedmannii* (17.9%), *Cystobasidium ongulense* (10.5%) and *Glaciozyma martinii* (10.5%) (Figure 3) [19].

Three genera and three species of ascomycetous fungi (i.e., *Phoma herbarum*, *Pseudogymnoascus pannorum*, and *Thelebolus microsporus*), and four genera and four species of basidiomycetous fungi (i.e., *M. gelida*, *N. friedmannii*, *Phenoliferia glacialis*, and *V. victoriae*), were common between the two sampling sites. These two areas are 60 km apart. However, the existing fungal diversity suggests that the fungal ecosystem near the Syowa Station was formed in a narrow space.

During the fungal surveys near the Syowa Station, two new fungal species, *Cystobasidium tubakii* and *C. ongulense*, were identified for the first time in JARE's his-

tory [20]. Also, the number of fungal species reported near the station increased from 16 to 77 species (nearly a fivefold increase) from 2013 to 2022 (Table 1) [21].

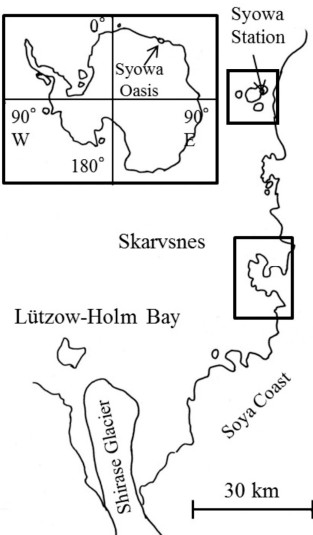

**Figure 1.** Map of Antarctica and the locations of the Skarvsnes ice-free area and Syowa Station (East Ongul Island).

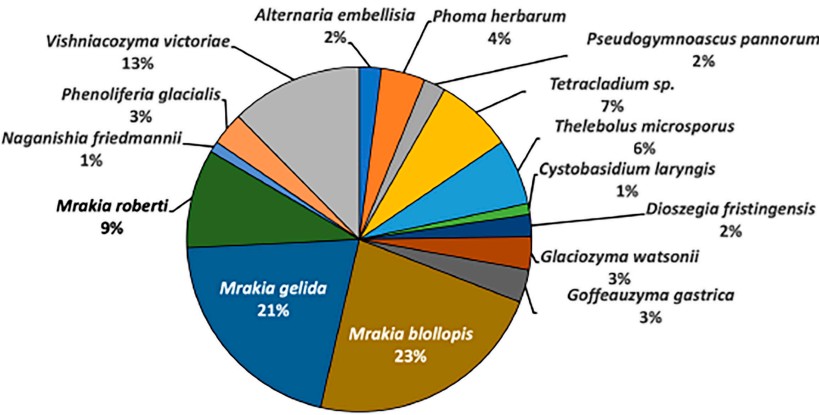

**Figure 2.** Fungal diversities in the Skarvsnes ice-free area.

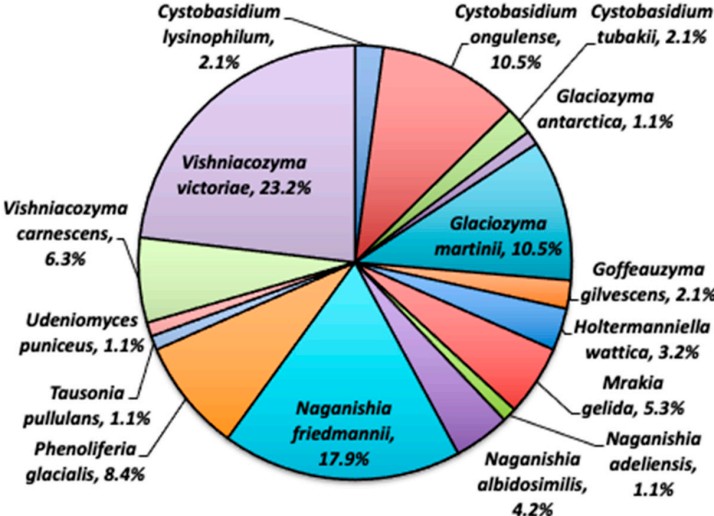

**Figure 3.** Fungal diversities on the East Ongul island.

## 4. Fungal Diversity Research in the Glacier Retreat Area of Svalbard, High Arctic

Austre Brøggerbreen (79° N, 12° E) is located in Ny-Ålesund in the Svalbard archipelago, Norway. The glacier has been markedly affected by climate change and is one of the most retreating glaciers in the world [22]. Therefore, this review examines the fungal changes over time.

This study included five sites: Site 0 was upstream of the glacier (seven meters upstream of the glacier terminus), whereas Sites 1–4 were in the deglaciation area.

At Site 0, only isolated Basidiomycete yeasts (replaced by the genus/species) were found. At Site 1 (approximately ten years after the glacier retreated), one zygomycete (replaced by the genus/species) was isolated. At Site 2 (approximately 50 years after the glacier retreated), the genus Mrakia and a zygomycete (replaced by the genus/species) were isolated. At Site 3 (approximately 80 years after the glacier retreated), a zygomycete (replaced by the genus/species), ascomycete (replace by the genus/species), and basidiomycete yeast (replaced by the genus/species) were isolated (Figure 4).

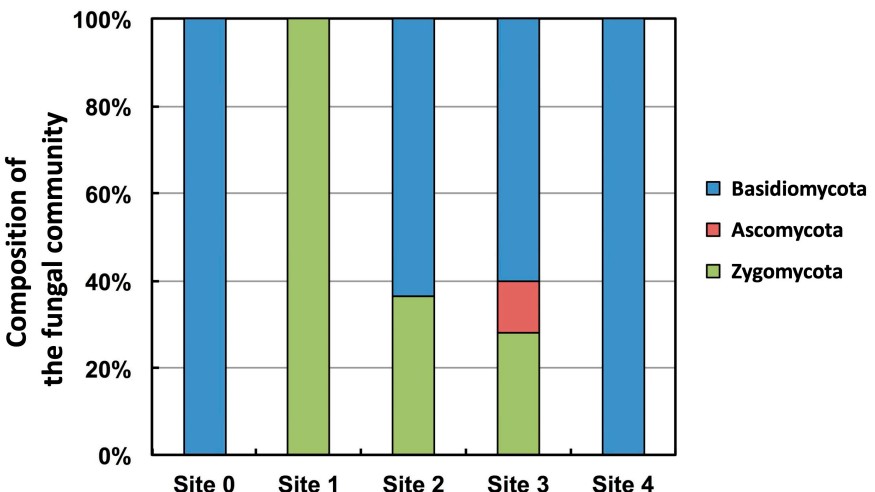

**Figure 4.** Composition of the fungal community at each sampling point in the Austre Brøggerbreen retreat area.

Based on these results, the changes in the fungal community in the Austre Brøggerbreen retreat area process can be inferred as follows.

1.  Immediately after the glacier retreated and exposed the ground, *Mortierella* spp. and *Mucor* spp., both of which belong to Zygomycota, colonized the area (Site 1).
2.  *Mrakia* sp. colonized the area by utilizing the nutrients produced by these zygomycetes (Site 2).
3.  To colonize the area, ascomycete and basidiomycete yeasts (except for *Mrakia* sp.) utilized the nutrients accumulated by *Mrakia* sp. (Site 3).

Thus, the diversity of fungi in the Austre Brøggerbreen retreat area has changed significantly for more than 80 years [23].

## 5. Fungal Survey in the Ellesmere Island, Canadian High Arctic

Ellesmere Island in Canada is known as the northernmost inhabited island in the world. Fungal surveys were conducted in the northern part of Ellesmere Island on Ice Island (82°50′ N, 73° 40′ W) and the Walker Glacier (unofficial name: 83°00′ N, 72°12′ W) (Figure 5).

The Walker Glacier has experienced a retreating terminus at an average rate of 1.3 m/y from 1959 to 2013 due to glacial melting, thus exposing the ground. However, from 2013 to 2016, the glacier retreat rate increased to 3.3 m/y, suggesting that significant climate change has also occurred in the High Arctic. In this glacial retreat area, fungi were cultured

at nine sites, wherein two were isolated from the glacier and seven were isolated from the retreat area. Changes in the fungal community at each site were examined.

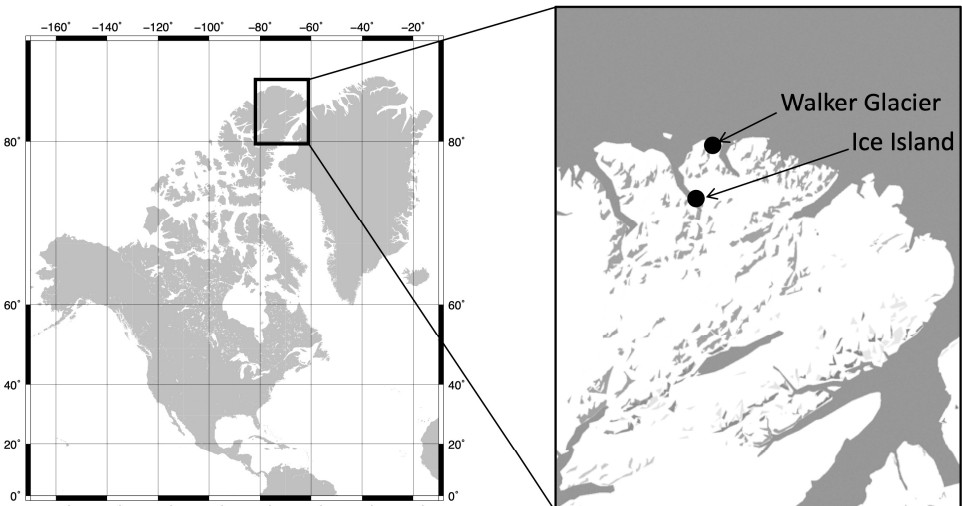

**Figure 5.** Map of the Canadian High Arctic.

A total of 325 fungal strains were isolated, of which the DNA (ITS and D1/D2 regions) of 275 strains was successfully sequenced. These strains were classified into 11 genera and 11 species of Ascomycota, 7 genera and 12 species of Basidiomycota, and 2 genera and 6 species of Zygomycota.

Cluster analysis showed that the fungal community was divided into two groups: on glaciers and in glacial retreat areas (Figure 6). Several species were yet to be reported among the fungi living on the Walker Glacier. These novel species may be endemic to this region, indicating that if climate change continues and the glaciers are entirely lost, many fungi living on the glaciers will lose their habitat and become extinct. Thus, global warming in the Arctic region affects not only animals, such as polar bears, but microorganisms, such as fungi [24].

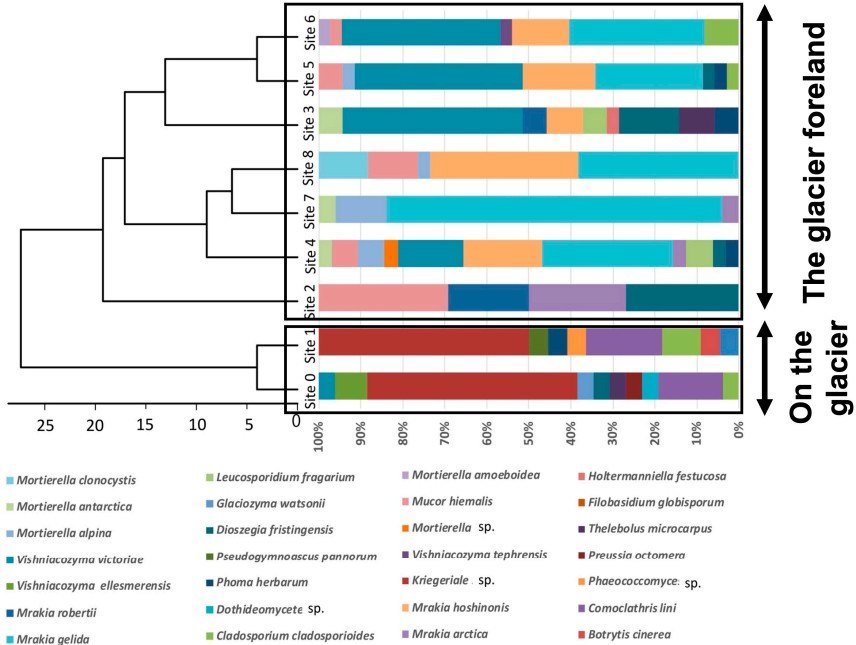

**Figure 6.** Dendrogram of the fungal communities among the nine sites in the Walker Glacier and its foreland.

During these fungal surveys, basidiomycete yeasts were reported as new fungal species: *Gelidatrema psychrophila* and *Mrakia arctica* from the Ice Island, and *Vishniacozyma ellesmerensis* and *Mrakia hoshinonis* from the Walker Glacier [25–28].

## 6. Growth and Enzyme Activities at Sub-Zero Temperatures

Fungi cannot regulate their intracellular temperature. Thus, the extracellular environment, such as external and water temperatures, affects their biological activities. Therefore, it is difficult for fungi to maintain their vital activities in extreme environments, such as the Antarctic and Arctic regions, due to intracellular freezing and reduced enzyme activity caused by low temperatures. In the vicinity of Syowa Station and Ellesmere Island, the average maximum temperature is below 0 °C for most days of the year. However, 77 fungal species have been reported from the Syowa Station area, whereas more than 30 have been isolated from Ellesmere Island.

Fungi living in the polar regions were examined for their activity in sub-zero environments. Nineteen species of fungi isolated from the East Ongul Island and Ellesmere Island were selected and cultured at −3 °C to analyze their ability to grow at sub-zero temperatures (below −3 °C YPD liquid medium or potato dextrose agar medium freeze). The results confirmed growth at −3 °C for all species, regardless of the habitat, optimal growth temperature, or maximum growth temperature (Table 2) [24,26,27]. Furthermore, fungi living in the Antarctic and Arctic regions can degrade extracellular polymers even at −3 °C and utilize these compounds for growth, indicating that these fungi harbor enzymes that are active at −3 °C and secrete these enzymes extracellularly. The ability of the same strains to secrete enzymes at −3 °C was examined and calculated using the following formula:

Extracellular enzyme secretion ability = (clear or opaque zone diameter—colony diameter)/colony diameter

The secretion ability was assessed as follows: ++, strongly positive, for values > 2.0; +, positive, for values between 1.0 and 2.0; w, weakly positive, for values < 1.0; and –, negative, no clear zone.

**Table 2.** Ability of fungi isolated from the East Ongul Island in Antarctica and the Ellesmere Island in the Canadian High Arctic to grow.

| Species | Habitat | Growth at −3 °C | Optimum Growth Temperature | Maximum Growth Temperature |
|---|---|---|---|---|
| *Cystobasidium lysinophilum* | East Ongul Island | + | 25 °C | 30 °C |
| *Cystobasidium ongulense* | East Ongul Island | + | 20 °C | 30 °C |
| *Cystobasidium tubakii* | East Ongul Island | + | 15–17 °C | 25 °C |
| *Glaciozyma Antarctica* | East Ongul Island | + | 10 °C | 15 °C |
| *Glaciozyma martinii* | East Ongul Island | + | 15 °C | 17 °C |
| *Goffeauzyma gilvescens* | East Ongul Island | + | 20 °C | 25 °C |
| *Holtermanniella wattica* | East Ongul Island | + | 15 °C | 25 °C |
| *Mrakia arctica* | Ellesmere Island | + | 15 °C | 20 °C |
| *Mrakia gelida* | East Ongul Island | + | 15 °C | 20 °C |
| *Mrakia hoshinonis* | Ellesmere Island | + | 15 °C | 20 °C |
| *Naganishia adeliensis* | East Ongul Island | + | 25 °C | 30 °C |
| *Naganishia albidosimilis* | East Ongul Island | + | 25 °C | 30 °C |
| *Naganishia friedmannii* | East Ongul Island | + | 20 °C | 25 °C |
| *Phenoliferia glacialis* | East Ongul Island | + | 15 °C | 17 °C |
| *Tausonia pullulans* | East Ongul Island | + | 15 °C | 25 °C |
| *Udeniomyces puniceus* | East Ongul Island | + | 20 °C | 25 °C |
| *Vishniacozyma carnescens* | East Ongul Island | + | 20 °C | 25 °C |
| *Vishniacozyma ellesmerensis* | Ellesmere Island | + | 15–17 °C | 20 °C |
| *Vishniacozyma victoriae* | East Ongul Island | + | 17 °C | 25 °C |

Culture experiments were performed on potato dextrose agar medium. + indicates evident growth within seven days after culture.

Table 3 summarizes the results of the enzyme secretion tests for lipase, cellulase, and protease from the fungal strains isolated from East Ongul Island and Ellesmere Island.

*Holtermanniella wattica*, *Tausonia pullulans*, and *Goffeauzyma gilvescens* isolated from East Ongul Island, and *M. arctica* and *M. hoshinonis* isolated from Ellesmere Island, showed a high ability to secrete lipase at −3 °C. Meanwhile, *Udeniomyces puniceus* isolated from East Ongul Island, and *M. arctica* and *M. hoshinonis* isolated from Ellesmere Island, showed a higher ability to secrete cellulase at −3 °C [25,27,28]. None of the fungi isolated from the Antarctic and Arctic regions showed a high ability to secrete protease at −3 °C. It

is noteworthy that *M. arctica* and *M. hoshinonis* isolated from Ellesmere Island showed a higher ability to secrete cellulase and lipase than the fungal strains from Antarctica [25,28].

These results indicate that fungi in the Antarctic and Arctic regions can secrete active enzymes even at sub-zero temperatures, suggesting that these fungi play an essential role in decomposing organic materials in environments where the temperature is below freezing most days of the year, such as in polar regions.

**Table 3.** Ability of fungi isolated from the Antarctic and Canadian High Arctic to secrete enzymes at $-3\,^{\circ}\text{C}$.

| Species | Habitat | Lipase | Cellulase | Protease |
|---|---|---|---|---|
| *Cystobasidium lysinophilum* | East Ongul Island | - | - | - |
| *Cystobasidium ongulense* | East Ongul Island | 0.43 ± 0.05 | 0.17 ± 0.06 | - |
| *Cystobasidium tubakii* | East Ongul Island | 0.19 ± 0.07 | 0.13 ± 0.01 | - |
| *Glaciozyma Antarctica* | East Ongul Island | 0.41 ± 0.13 | - | - |
| *Glaciozyma martinii* | East Ongul Island | - | - | - |
| *Goffeauzyma gilvescens* | East Ongul Island | 2.50 ± 0.20 | - | - |
| *Holtermanniella wattica* | East Ongul Island | 2.27 ± 0.09 | - | - |
| *Mrakia arctica* | Ellesmere Island | 6.15 ± 0.68 | 5.34 ± 0.78 | 0.75 ± 0.12 |
| *Mrakia gelida* | East Ongul Island | - | 0.35 ± 0.07 | - |
| *Mrakia hoshinonis* | Ellesmere Island | 4.29 ± 0.34 | 2.55 ± 0.20 | 1.53 ± 0.05 |
| *Naganishia adeliensis* | East Ongul Island | 1.23 ± 0.41 | - | 0.40 ± 0.12 |
| *Naganishia albidosimilis* | East Ongul Island | 0.84 ± 0.16 | - | - |
| *Naganishia friedmannii* | East Ongul Island | - | - | 1.11 ± 0.08 |
| *Phenoliferia glacialis* | East Ongul Island | - | - | - |
| *Tausonia pullulans* | East Ongul Island | 2.60 ± 0.33 | 1.88 ± 0.21 | - |
| *Udeniomyces puniceus* | East Ongul Island | - | 2.92 ± 0.37 | 0.86 ± 0.35 |
| *Vishniacozyma carnescens* | East Ongul Island | 0.56 ± 0.31 | 0.49 ± 0.22 | - |
| *Vishniacozyma ellesmerensis* | Ellesmere Island | 1.56 ± 0.16 | - | - |
| *Vishniacozyma victoriae* | East Ongul Island | 0.62 ± 0.03 | - | 0.49 ± 0.18 |

The values represent the difference between the diameters of the zone of clearance and the colony, expressed as a proportion of the colony size (means ± SD for triplicates). -; no activity.

## 7. Future Research Prospects

Since fungal surveys conducted through culture methods have inherent limitations in terms of the number of fungal species and strains that can be successfully cultured, the utilization of next-generation sequencers for the diversity analysis of fungi in polar regions has garnered significant attention in recent years. The Illumina platforms Miseq and Hiseq, commonly used for microbial diversity analysis, have relatively short read lengths of 150–300 base pairs. Due to this short sequence length, analyzing fungal diversity, even at the genus level, can be challenging. To compensate for the short read lengths of Miseq and Hiseq, a long-read next-generation sequencer, such as MinION and PacBio Sequel2 sequencing, can be used to conduct a comprehensive fungal study at the species level. Fungal surveys have yet to be undertaken in many areas near Syowa Station in Antarctica and Ellesmere Island in the Canadian High Arctic. The investigation of these areas' fungal diversity and climate change's effects on the fungal community should be continued.

Because Antarctic and Arctic fungal strains are at risk of habitat loss and extinction due to climate change, systematic preservation for the future generations of these strains should be attempted. In addition, the genome sequence of Antarctic fungi has attracted attention as a novel genetic resource because it provides information on active enzymes even at sub-zero temperatures and new pharmaceutical raw materials.

The initial draft genome sequence of *Mrakia blollopis* was the world's first Antarctic fungus to be reported [29]. Following this, the genome sequences of other Antarctic fungi, including *Cystobasidium tubakii*, *C. ongulense*, *M. gelida*, and several others, have been deposited in DNA databases such as NCBI/DDBJ/EMBL [30–32].

By predicting gene sequences from the whole genome sequences of Antarctic fungi and publishing the results in a genome database, I would like to make genome infor-

mation on Antarctic fungi available to everyone, even if they need to gain knowledge on bioinformatics.

Universities, research institutes, and private companies should actively focus on Antarctic fungi, allowing Japan to lead the world in establishing a new research field using Antarctic fungi.

**Funding:** This work was supported by the NIPR Research Project (KP-309), a JSPS Grant-in-Aid for Scientific Research(B), granted to M. Tsuji (no. 23H03590), Institution for Fermentation, Osaka, the General Research Grant, granted to M. Tsuji (no. G-2022-1-007), and the ArCS2 (Arctic Challenge for Sustainability) II project (Program Grant Number JPMXD1420318865), provided by the Ministry of Education, Culture, Sports, Science and Technology, Japan. This work was also supported by the "Strategic Research Projects" grant from ROIS (Research Organization of Information and Systems) (no. 2022-SRP-01).

**Data Availability Statement:** Not applicable.

**Acknowledgments:** The author would like to thank the National Institute of Polar Research, Japan, and The National Institute of Technology, Asahikawa College, Japan, for their support in obtaining these research results. This work contributes to the National Institute of Polar Research, Japan mission and the International Arctic Science Committee (IASC) project T-MOSAiC (Terrestrial Multidisciplinary distributed Observatories for the Study of Arctic Connections).

**Conflicts of Interest:** The authors declares no conflict of interest.

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
