# Peer review of "Survey on Fungi in Antarctica and High Arctic Regions, and Their Impact on Climate Change"

_climate, doi:10.3390/cli11090195_

Round 1

Reviewer 1 Report

Dear author, the purpose of the review has merit and is very well executed, with very up-to-date and appropriate procedures and technologies. I only have a coment regarding the first paragraph of the introduction, line 20-21 i think this line can be improved. Kind regards

Dear author, the purpose of the review has merit and is very well executed, with very up-to-date and appropriate procedures and technologies. I only have a coment regarding the first paragraph of the introduction, line 20-21 i think this line can be improved. Kind regards

Author Response

Thank you for your valiable comment.

According to your comment, I have revised in the first paragraph of introduction; new text line21-23. 

Reviewer 2 Report

The topic is extremely relevant, but the article does not fulfill the purpose of a “review” article and therefore should not be published in this format that it was written. The author insufficiently cites the work of other research groups, leaving the discussion poor and superficial. In addition, the number of self-citations is very high.

Here, it is the link of book on Antarctic fungi, published in 2019: https://link.springer.com/book/10.1007/978-3-030-18367-7

The introduction, for example, should be rewritten and supported by scientific references about the subject. There is no reference in the first paragraph of introduction. There are only two references in this part and the third paragraph (Line 30) is too short. Line 140 is another example of a super short paragraph.

The article discusses the survival of fungi at the poles and the production of enzymes at low temperatures. Have these experiments been published yet? If so, what is the importance of re-presenting these data in a simplified way in a review article?

And finally, the author discussed the availability of the genome of fungi obtained from the mentioned ecosystems as a perspective, although he does not deal about the diversity of fungi that cannot be cultivated and he does not present any result about fungi from metabarcoding samples. 

Writing is clear and quality of english language is ok.

Author Response

I appreciate your thought provoking comments.

Following comments from other reviewers, the manuscript was substantially revised.

For the experiments with enzymes outside the fungus, we use data that have already been published.
While there are many data available on exoenzyme experiments on low-temperature microorganisms, there are fewer available on Antarctic and Arctic fungi.

As you know, Miseq and Hiseq cannot provide reliable data due to their short read lengths.
There are a few papers available, but those that analyze at the genus level are not reliable.
The analysis of fungal diversity in polar regions using NGS is still a developing field and is one of our future tasks.

Reviewer 3 Report

Title: Survey on fungi in Antarctica and High Arctic regions, and their impact on climate change

Reference: climate-2532021

Type: Review

Overview:

The manuscript climate-2532021 entitled “Survey on fungi in Antarctica and High Arctic regions, and their impact on climate change” intends to summarize the changes in fungal communities associated with climate change, based on previously published data.

General comments:

Changes to the manuscript are required. First and foremost, the aims should be further stressed. Moreover, adaptations to formal writing (written in the third person, please consider rewriting sentences to remove the personal pronoun I) must be contemplated. Also, the lack of consistency (e.g., British and American spelling; subzero and sub-zero; retreat and receded) should be addressed.

Finally, the manuscript is quite descriptive. It would be interesting to extract general trends (passing concrete relevant information into tables) and comment on them.

Specific comments:

Abstract: Please consider replacing with (changes presented in bold):

Abstract: The Antarctica and High Arctic regions are extreme environments, with average maximum temperatures below 0 °C for most days of the year. Interestingly, fungi inhabit these regions. This review describes the history of fungal surveys near the Syowa Station and the fungal diversity in this region. In the High Arctic region, I summarized the changes in fungal communities in glacial retreat areas of Ny-Ålesund, Norway, and Ellesmere Island, Canada, in response to climate change. In addition, the growth and enzyme secretion ability of Antarctic and Arctic fungi at sub-zero temperatures is presented. Finally, I have also provided the future directions of Antarctic and Arctic fungal research.

Keywords: Please consider organizing by alphabetic order. Please consider adding at least one more keyword.

Antarctica; climate change; fungi, High Arctic.

Introduction: Please consider replacing with:

The Arctic region is mainly occupied by the Arctic Ocean, which is surrounded by land. On the other hand, the Antarctic region is dominated by the Antarctic Continent, with ice and snow covering the continent. Together, the Arctic and Antarctic regions are called the Polar Regions.

Antarctica is one of the planet's extreme environments. It is exposed to cold and dry conditions, with the lowest temperature at –90 °C. Approximately 98% of its surface is covered by snow and ice, and the remaining 2% represents the ice-free area located in coastal areas and high mountains, wherein the snow melts in summer, exposing the ground. Most of the terrestrial ecosystems of Antarctica are distributed in this ice-free area [1]. Syowa Station, the headquarters for the Japanese Antarctic Research Expedition (JARE), is also an ice-free area.

The High Arctic, like Antarctica, is one of the most extreme environments, with a maximum temperature of below 0 °C for most of the year.

Despite its exposure to adverse conditions, such as subzero temperatures and low nutrient and water availability, the fungi that inhabit cold environments can grow at near-subzero temperatures. Secretion of extracellular enzymes allows them to utilize complex materials as energy sources. Therefore, psychrophilic and psychrotolerant fungi play essential roles in the nutrient cycle of the polar region ecosystems [2].

This review presents a history of fungal research in Antarctica, followed by a report on fungal surveys near the Syowa Station. In addition, I summarize the impact of climate changes on fungal diversity based on fungal surveys in glacier retreat areas in the High Arctic. I also introduce fungi's growth and enzyme activity inhabiting the polar regions at subzero temperatures. Finally, I discuss the prospects of fungal research in the Arctic and Antarctic regions.

Development: Please consider replacing with:

2. History of fungal research in the Antarctic region and near Syowa Station

Antarctica was sighted in 1820, and the first landing occurred a year later [ref missing]. The first report on fungi from this region was published circa 1897–1899 by a Belgian expedition that collected Sclerotium antarcticum from Danco Island near the Antarctic Peninsula [3]. 

In the 1960s, several new fungal species were reported by Fell et al. [4] and di Menna [5-6]. Bridge and Spooner (2012) created a "list of non-lichenized fungi from the Antarctic region", an essential tool to understand Antarctica's fungal diversity. However, the list still needs to be updated [7].

The first report on fungi around Syowa Station, the headquarters of the Japanese Antarctic Research Expedition (JARE), was published in 1961 [8], followed by three articles that reported 12 ascomycetes and four basidiomycetes (Table 1) [9-11].

In the 1960s, the JARE was suspended. Nevertheless, during this period, Moesziomyces antarcticus was isolated from Ross Island [12]. This fungal species secretes stable lipase and produces biosurfactants [13-14].

From the 1960s to 2013, no additional fungal species were reported near the Syowa Station by JARE. After this date, the JARE activities revealed … (complete the sentence) [15-16]).

3. Fungal diversity research near the Syowa Station

Samples collected during JARE–48 and JARE–49 allowed the survey of the fungal diversity from the Skarvsnes ice-free area and East Ongul Island (Figure 1). In the Skarvsnes ice-free area, 97 fungal strains were isolated and classified into five genera and five species of Ascomycota, and eight genera and ten species of Basidiomycota. Mrakia spp. was the dominant genus (53%), corresponding to the species M. blollopis (23%), M. gelida (21%), and M. robertii (9%) (Figure 2) [15]. The genus Mrakia is a basidiomycete yeast previously isolated from various cold environments worldwide, including the Arctic region, the Himalayas, the Alps, and Antarctica [17].

On East Ongul Island, 196 fungal strains were isolated and classified into eight genera and nine species of Ascomycota, and ten genera and 16 species of Basidiomycota. The dominant fungal species on the island were Thelebolus microspores (27%), Vishniacozyma victoriae (11%), and Naganishia friedmannii (9%) (Figure 3) [18].

Three genera and three species of ascomycetous fungi (i.e., Phoma herbarum, Pseudogymnoascus pannorum, and Thelebolus microsporus), and four genera and four species of basidiomycetous fungi (i.e., M. gelida, N. friedmannii, Phenoliferia glacialis, and V. victoriae) were common between the two sampling sites. These two areas are 60 km apart. However, the existing fungal diversity suggests that the fungal ecosystem near the Syowa Station was formed in a narrow space.

During the fungal surveys near the Syowa Station, two new fungal species, Cystobasidium tubakii and C. ongulense, were identified for the first time [19]. Also, the number of fungal species reported near the station increased from 16 to 77 species (nearly a fivefold increase) from 2013 to 2022 (Table 1) [20].

4. Fungal diversity research in the glacier retreat area of Svalbard, High Arctic

Austre Brøggerbreen (79°N, 12°E) is in Ny-Ålesund in the Svalbard archipelago, Norway. The glacier has been markedly affected by climate change and is one of the most retreating glaciers in the world [21]. Therefore, this review examines the fungal changes over time. This study included five sites: Site 0 was upstream of the glacier (seven meters upstream of the glacier terminus), whereas Sites 1–4 were in the deglaciation area.

Site 0, were only isolated Basidiomycete yeasts (replace by the genus/species). At Site 1 (approximately ten years after the glacier retreated), one zygomycete (replace by the genus/species) was isolated. At Site 2 (approximately 50 years after the glacier retreated), the genus Mrakia and a zygomycete (replace by the genus/species) were isolated. At Site 3 (approximately 80 years after the glacier retreated), a zygomycete (replace by the genus/species), ascomycete (replace by the genus/species), and basidiomycete yeast (replace by the genus/species) were isolated (Figure 4).

Based on these results, the changes in the fungal community in the Austre Brøggerbreen retreat area process can be inferred as follows:

1. Immediately after the glacier retreated and exposed the ground, Mortierella spp. and Mucor spp. (Zygomycota) colonized the area (Site 1).

2. Mrakia sp. colonized the area by utilizing the nutrients produced by these zygomycetes (Site 2).

3. To colonize the area, ascomycete and basidiomycete yeasts (except for Mrakia sp.) utilized the nutrients accumulated by Mrakia sp. (Site 3).

Thus, the diversity of fungi in the Austre Brøggerbreen retreat area has changed significantly for more than 80 years [22].

5. Fungal survey in the Ellesmere Island, Canadian High Arctic

Ellesmere Island, in Canada, is known as the northernmost inhabited island in the world. Fungal surveys were conducted in the northern part of Ellesmere Island on Ice Island (82°50′ N, 73° 40′ W) and Walker Glacier (unofficial name: 83°00′ N, 72°12′ W) (Figure 5). 

The Walker Glacier had been retreating its terminus at an average rate of 1.3 m/y from 1959 to 2013 due to glacial melting, thus exposing the ground. However, from 2013 to 2016, the glacier retreat rate increased to 3.3 m/y, suggesting that significant climate change also occurred in the High Arctic. In this glacial retreat area, fungi were cultured at nine sites; two were isolated from the glacier and seven from the retreat area.

Changes in the fungal community at each site were examined. A total of 325 fungal strains were isolated, of which the DNA (ITS and D1/D2 regions) of 275 strains were successfully sequenced. These strains were classified into 11 genera and 11 species of Ascomycota, seven genera and 12 species of Basidiomycota, and two genera and six species of Zygomycota. Cluster analysis showed that the fungal community was divided into two groups: on glaciers and in glacial retreat areas (Figure 6). Several species have yet to be previously reported among the fungi living on the Walker Glacier. These novel species may be endemic to this region, indicating that if climate change continues and the glaciers are entirely lost, many fungi living on the glaciers will lose their habitat and become extinct. Thus, global warming in the Arctic region affects not only animals, such as polar bears, but microorganisms, such as fungi [23].

During these fungal surveys, basidiomycete yeasts were reported as new fungal species: Gelidatrema psychrophile and Mrakia arctica from the Ice Island, and Vishniacozyma ellesmerensis and Mrakia hoshinonis from the Walker Glacier [24-27]. 

6. Growth and enzyme activities at sub-zero temperatures

Fungi cannot regulate their intracellular temperature. Thus, the extracellular environment, such as external and water temperatures, affects their biological activities. Therefore, it is difficult for fungi to maintain their vital activities in extreme environments, such as the Antarctic and Arctic regions, due to intracellular freezing and reduced enzyme activity caused by low temperatures. In the vicinity of Syowa Station and Ellesmere Island, the average maximum temperature is below 0℃ for most days of the year. However, 77 fungal species have been isolated from the Syowa Station area, whereas more than 30 were isolated from Ellesmere Island. 

Fungi living in the polar regions were examined for their activity in sub-zero environments. Nineteen species of fungi isolated from East Ongul Island and Ellesmere Island were selected and cultured at −3°C to analyze their ability to grow at sub-zero temperatures (below −3°C YPD liquid medium or potato dextrose agar medium freeze). Results confirmed growth at −3°C for all species, regardless of habitat, optimal growth temperature, or maximum growth temperature (Table 2) [24, 26-27]. Furthermore, fungi living in the Antarctic and Arctic regions can degrade extracellular polymers, even at −3°C, and utilize these compounds for growth, indicating that these fungi harbour enzymes that are active at −3°C and secrete these enzymes extracellularly. The ability of the same strains to secrete enzymes at −3°C was examined and calculated using the following formula:

Extracellular enzyme secretion ability = (clear or opaque zone diameter – colony diameter) / colony diameter

Secretion ability was assessed as follows: ++, strongly positive, for values >2.0; +, positive, for values between 1.0 and 2.0; w, weakly positive, for values < 1.0; and –, negative, no clear zone. 

Culture experiments were performed on potato dextrose agar medium. + indicates an evident growth within seven days after culture.

Table 3 summarizes the results of the enzyme secretion tests for lipase, cellulase, and protease from the fungal strains isolated from East Ongul Island and Ellesmere Island.

The values represent the difference between the diameters of the zone of clearance and the colony, expressed as a proportion of the colony size (means ± SD for triplicates).- ; no activity.

Holtermanniella watticaTausonia pullulans, and Goffeauzyma gilvescens isolated from East Ongul Island, and M. arctica and M. hoshinonis isolated from Ellesmere Island showed high lipase secretion ability at −3°C. Udeniomyces puniceus isolated from East Ongul Island, and M. arctica and M. hoshinonis isolated from Ellesmere Island showed high cellulase secretion ability at −3°C [24,26-27]. None of the fungi isolated from the Antarctic and Arctic regions showed high protease secretion ability at −3°C. NoteworthyM. arctica and M. hoshinonis isolated from Ellesmere Island showed higher secretory ability for cellulase and lipase than the fungal strains from Antarctica [24,27]. 

These results indicate that fungi in the Antarctic and Arctic regions can secrete active enzymes even at sub-zero temperatures, suggesting that these fungi play an essential role in decomposing organic materials in environments where the temperature is below freezing most days of the year, such as in polar regions.

7. Future research prospects 

In addition to the fungal survey using the culture-based method, a long-read next-generation sequencer, such as MinION and PacBio Sequel2 sequencing, can be used to conduct a comprehensive fungal study at the species level. Fungal surveys have yet to be undertaken in many areas near Syowa Station in Antarctica and Ellesmere Island in the Canadian High Arctic. Investigation of these areas' fungal diversity and climate change's effects on the fungal community should be continued. 

Because Antarctic and Arctic fungal strains are at risk of habitat loss and extinction due to climate change, systematic preservation for the future generations of these strains should be attempted. In addition, the genome sequence of Antarctic fungi has attracted attention as a novel genetic resource because it provides information on active enzymes even at sub-zero temperatures and new pharmaceutical raw materials. By predicting gene sequences from the whole genome sequences of Antarctic fungi and publishing the results in a genome database, I would like to make genome information on Antarctic fungi available to everyone, even if they need to gain knowledge on bioinformatics. 

Universities, research institutes, and private companies should actively focus on Antarctic fungi, allowing Japan to lead the world in establishing a new research field using Antarctic fungi.

Figures

Figure 6: Please clarify where sites 5-7 are located.

Scientific comments:

Line 62: Please consider further explaining the role of stable lipase and biosurfactants in fungal survival under adverse conditions.

Lines 63-66: please consider summarizing the information collected by Dr. Tamotsu Hoshino and Dr. Takashi Osono.

Line 98: further explain the sentence (“narrow space”?).

Lines 162-164: Please further explain (seems contradictory since there was a diversity gain – novel species appeared)

Lines 192-193: Insert as an equation (option on word)

Title: Survey on fungi in Antarctica and High Arctic regions, and their impact on climate change

Reference: climate-2532021

Type: Review

Overview:

The manuscript climate-2532021 entitled “Survey on fungi in Antarctica and High Arctic regions, and their impact on climate change” intends to summarize the changes in fungal communities associated with climate change, based on previously published data.

General comments:

Changes to the manuscript are required. First and foremost, the aims should be further stressed. Moreover, adaptations to formal writing (written in the third person, please consider rewriting sentences to remove the personal pronoun I) must be contemplated. Also, the lack of consistency (e.g., British and American spelling; subzero and sub-zero; retreat and receded) should be addressed.

Finally, the manuscript is quite descriptive. It would be interesting to extract general trends (passing concrete relevant information into tables) and comment on them.

Specific comments:

Abstract: Please consider replacing with (changes presented in bold):

Abstract: The Antarctica and High Arctic regions are extreme environments, with average maximum temperatures below 0 °C for most days of the year. Interestingly, fungi inhabit these regions. This review describes the history of fungal surveys near the Syowa Station and the fungal diversity in this region. In the High Arctic region, I summarized the changes in fungal communities in glacial retreat areas of Ny-Ålesund, Norway, and Ellesmere Island, Canada, in response to climate change. In addition, the growth and enzyme secretion ability of Antarctic and Arctic fungi at sub-zero temperatures is presented. Finally, I have also provided the future directions of Antarctic and Arctic fungal research.

Keywords: Please consider organizing by alphabetic order. Please consider adding at least one more keyword.

Antarctica; climate change; fungi, High Arctic.

Introduction: Please consider replacing with:

The Arctic region is mainly occupied by the Arctic Ocean, which is surrounded by land. On the other hand, the Antarctic region is dominated by the Antarctic Continent, with ice and snow covering the continent. Together, the Arctic and Antarctic regions are called the Polar Regions.

Antarctica is one of the planet's extreme environments. It is exposed to cold and dry conditions, with the lowest temperature at –90 °C. Approximately 98% of its surface is covered by snow and ice, and the remaining 2% represents the ice-free area located in coastal areas and high mountains, wherein the snow melts in summer, exposing the ground. Most of the terrestrial ecosystems of Antarctica are distributed in this ice-free area [1]. Syowa Station, the headquarters for the Japanese Antarctic Research Expedition (JARE), is also an ice-free area.

The High Arctic, like Antarctica, is one of the most extreme environments, with a maximum temperature of below 0 °C for most of the year.

Despite its exposure to adverse conditions, such as subzero temperatures and low nutrient and water availability, the fungi that inhabit cold environments can grow at near-subzero temperatures. Secretion of extracellular enzymes allows them to utilize complex materials as energy sources. Therefore, psychrophilic and psychrotolerant fungi play essential roles in the nutrient cycle of the polar region ecosystems [2].

This review presents a history of fungal research in Antarctica, followed by a report on fungal surveys near the Syowa Station. In addition, I summarize the impact of climate changes on fungal diversity based on fungal surveys in glacier retreat areas in the High Arctic. I also introduce fungi's growth and enzyme activity inhabiting the polar regions at subzero temperatures. Finally, I discuss the prospects of fungal research in the Arctic and Antarctic regions.

Development: Please consider replacing with:

2. History of fungal research in the Antarctic region and near Syowa Station

Antarctica was sighted in 1820, and the first landing occurred a year later [ref missing]. The first report on fungi from this region was published circa 1897–1899 by a Belgian expedition that collected Sclerotium antarcticum from Danco Island near the Antarctic Peninsula [3]. 

In the 1960s, several new fungal species were reported by Fell et al. [4] and di Menna [5-6]. Bridge and Spooner (2012) created a "list of non-lichenized fungi from the Antarctic region", an essential tool to understand Antarctica's fungal diversity. However, the list still needs to be updated [7].

The first report on fungi around Syowa Station, the headquarters of the Japanese Antarctic Research Expedition (JARE), was published in 1961 [8], followed by three articles that reported 12 ascomycetes and four basidiomycetes (Table 1) [9-11].

In the 1960s, the JARE was suspended. Nevertheless, during this period, Moesziomyces antarcticus was isolated from Ross Island [12]. This fungal species secretes stable lipase and produces biosurfactants [13-14].

From the 1960s to 2013, no additional fungal species were reported near the Syowa Station by JARE. After this date, the JARE activities revealed … (complete the sentence) [15-16]).

3. Fungal diversity research near the Syowa Station

Samples collected during JARE–48 and JARE–49 allowed the survey of the fungal diversity from the Skarvsnes ice-free area and East Ongul Island (Figure 1). In the Skarvsnes ice-free area, 97 fungal strains were isolated and classified into five genera and five species of Ascomycota, and eight genera and ten species of Basidiomycota. Mrakia spp. was the dominant genus (53%), corresponding to the species M. blollopis (23%), M. gelida (21%), and M. robertii (9%) (Figure 2) [15]. The genus Mrakia is a basidiomycete yeast previously isolated from various cold environments worldwide, including the Arctic region, the Himalayas, the Alps, and Antarctica [17].

On East Ongul Island, 196 fungal strains were isolated and classified into eight genera and nine species of Ascomycota, and ten genera and 16 species of Basidiomycota. The dominant fungal species on the island were Thelebolus microspores (27%), Vishniacozyma victoriae (11%), and Naganishia friedmannii (9%) (Figure 3) [18].

Three genera and three species of ascomycetous fungi (i.e., Phoma herbarum, Pseudogymnoascus pannorum, and Thelebolus microsporus), and four genera and four species of basidiomycetous fungi (i.e., M. gelida, N. friedmannii, Phenoliferia glacialis, and V. victoriae) were common between the two sampling sites. These two areas are 60 km apart. However, the existing fungal diversity suggests that the fungal ecosystem near the Syowa Station was formed in a narrow space.

During the fungal surveys near the Syowa Station, two new fungal species, Cystobasidium tubakii and C. ongulense, were identified for the first time [19]. Also, the number of fungal species reported near the station increased from 16 to 77 species (nearly a fivefold increase) from 2013 to 2022 (Table 1) [20].

4. Fungal diversity research in the glacier retreat area of Svalbard, High Arctic

Austre Brøggerbreen (79°N, 12°E) is in Ny-Ålesund in the Svalbard archipelago, Norway. The glacier has been markedly affected by climate change and is one of the most retreating glaciers in the world [21]. Therefore, this review examines the fungal changes over time. This study included five sites: Site 0 was upstream of the glacier (seven meters upstream of the glacier terminus), whereas Sites 1–4 were in the deglaciation area.

Site 0, were only isolated Basidiomycete yeasts (replace by the genus/species). At Site 1 (approximately ten years after the glacier retreated), one zygomycete (replace by the genus/species) was isolated. At Site 2 (approximately 50 years after the glacier retreated), the genus Mrakia and a zygomycete (replace by the genus/species) were isolated. At Site 3 (approximately 80 years after the glacier retreated), a zygomycete (replace by the genus/species), ascomycete (replace by the genus/species), and basidiomycete yeast (replace by the genus/species) were isolated (Figure 4).

Based on these results, the changes in the fungal community in the Austre Brøggerbreen retreat area process can be inferred as follows:

1. Immediately after the glacier retreated and exposed the ground, Mortierella spp. and Mucor spp. (Zygomycota) colonized the area (Site 1).

2. Mrakia sp. colonized the area by utilizing the nutrients produced by these zygomycetes (Site 2).

3. To colonize the area, ascomycete and basidiomycete yeasts (except for Mrakia sp.) utilized the nutrients accumulated by Mrakia sp. (Site 3).

Thus, the diversity of fungi in the Austre Brøggerbreen retreat area has changed significantly for more than 80 years [22].

5. Fungal survey in the Ellesmere Island, Canadian High Arctic

Ellesmere Island, in Canada, is known as the northernmost inhabited island in the world. Fungal surveys were conducted in the northern part of Ellesmere Island on Ice Island (82°50′ N, 73° 40′ W) and Walker Glacier (unofficial name: 83°00′ N, 72°12′ W) (Figure 5). 

The Walker Glacier had been retreating its terminus at an average rate of 1.3 m/y from 1959 to 2013 due to glacial melting, thus exposing the ground. However, from 2013 to 2016, the glacier retreat rate increased to 3.3 m/y, suggesting that significant climate change also occurred in the High Arctic. In this glacial retreat area, fungi were cultured at nine sites; two were isolated from the glacier and seven from the retreat area.

Changes in the fungal community at each site were examined. A total of 325 fungal strains were isolated, of which the DNA (ITS and D1/D2 regions) of 275 strains were successfully sequenced. These strains were classified into 11 genera and 11 species of Ascomycota, seven genera and 12 species of Basidiomycota, and two genera and six species of Zygomycota. Cluster analysis showed that the fungal community was divided into two groups: on glaciers and in glacial retreat areas (Figure 6). Several species have yet to be previously reported among the fungi living on the Walker Glacier. These novel species may be endemic to this region, indicating that if climate change continues and the glaciers are entirely lost, many fungi living on the glaciers will lose their habitat and become extinct. Thus, global warming in the Arctic region affects not only animals, such as polar bears, but microorganisms, such as fungi [23].

During these fungal surveys, basidiomycete yeasts were reported as new fungal species: Gelidatrema psychrophile and Mrakia arctica from the Ice Island, and Vishniacozyma ellesmerensis and Mrakia hoshinonis from the Walker Glacier [24-27]. 

6. Growth and enzyme activities at sub-zero temperatures

Fungi cannot regulate their intracellular temperature. Thus, the extracellular environment, such as external and water temperatures, affects their biological activities. Therefore, it is difficult for fungi to maintain their vital activities in extreme environments, such as the Antarctic and Arctic regions, due to intracellular freezing and reduced enzyme activity caused by low temperatures. In the vicinity of Syowa Station and Ellesmere Island, the average maximum temperature is below 0℃ for most days of the year. However, 77 fungal species have been isolated from the Syowa Station area, whereas more than 30 were isolated from Ellesmere Island. 

Fungi living in the polar regions were examined for their activity in sub-zero environments. Nineteen species of fungi isolated from East Ongul Island and Ellesmere Island were selected and cultured at −3°C to analyze their ability to grow at sub-zero temperatures (below −3°C YPD liquid medium or potato dextrose agar medium freeze). Results confirmed growth at −3°C for all species, regardless of habitat, optimal growth temperature, or maximum growth temperature (Table 2) [24, 26-27]. Furthermore, fungi living in the Antarctic and Arctic regions can degrade extracellular polymers, even at −3°C, and utilize these compounds for growth, indicating that these fungi harbour enzymes that are active at −3°C and secrete these enzymes extracellularly. The ability of the same strains to secrete enzymes at −3°C was examined and calculated using the following formula:

Extracellular enzyme secretion ability = (clear or opaque zone diameter – colony diameter) / colony diameter

Secretion ability was assessed as follows: ++, strongly positive, for values >2.0; +, positive, for values between 1.0 and 2.0; w, weakly positive, for values < 1.0; and –, negative, no clear zone. 

Culture experiments were performed on potato dextrose agar medium. + indicates an evident growth within seven days after culture.

Table 3 summarizes the results of the enzyme secretion tests for lipase, cellulase, and protease from the fungal strains isolated from East Ongul Island and Ellesmere Island.

The values represent the difference between the diameters of the zone of clearance and the colony, expressed as a proportion of the colony size (means ± SD for triplicates).- ; no activity.

Holtermanniella watticaTausonia pullulans, and Goffeauzyma gilvescens isolated from East Ongul Island, and M. arctica and M. hoshinonis isolated from Ellesmere Island showed high lipase secretion ability at −3°C. Udeniomyces puniceus isolated from East Ongul Island, and M. arctica and M. hoshinonis isolated from Ellesmere Island showed high cellulase secretion ability at −3°C [24,26-27]. None of the fungi isolated from the Antarctic and Arctic regions showed high protease secretion ability at −3°C. NoteworthyM. arctica and M. hoshinonis isolated from Ellesmere Island showed higher secretory ability for cellulase and lipase than the fungal strains from Antarctica [24,27]. 

These results indicate that fungi in the Antarctic and Arctic regions can secrete active enzymes even at sub-zero temperatures, suggesting that these fungi play an essential role in decomposing organic materials in environments where the temperature is below freezing most days of the year, such as in polar regions.

7. Future research prospects 

In addition to the fungal survey using the culture-based method, a long-read next-generation sequencer, such as MinION and PacBio Sequel2 sequencing, can be used to conduct a comprehensive fungal study at the species level. Fungal surveys have yet to be undertaken in many areas near Syowa Station in Antarctica and Ellesmere Island in the Canadian High Arctic. Investigation of these areas' fungal diversity and climate change's effects on the fungal community should be continued. 

Because Antarctic and Arctic fungal strains are at risk of habitat loss and extinction due to climate change, systematic preservation for the future generations of these strains should be attempted. In addition, the genome sequence of Antarctic fungi has attracted attention as a novel genetic resource because it provides information on active enzymes even at sub-zero temperatures and new pharmaceutical raw materials. By predicting gene sequences from the whole genome sequences of Antarctic fungi and publishing the results in a genome database, I would like to make genome information on Antarctic fungi available to everyone, even if they need to gain knowledge on bioinformatics. 

Universities, research institutes, and private companies should actively focus on Antarctic fungi, allowing Japan to lead the world in establishing a new research field using Antarctic fungi.

Figures

Figure 6: Please clarify where sites 5-7 are located.

Scientific comments:

Line 62: Please consider further explaining the role of stable lipase and biosurfactants in fungal survival under adverse conditions.

Lines 63-66: please consider summarizing the information collected by Dr. Tamotsu Hoshino and Dr. Takashi Osono.

Line 98: further explain the sentence (“narrow space”?).

Lines 162-164: Please further explain (seems contradictory since there was a diversity gain – novel species appeared)

Lines 192-193: Insert as an equation (option on word)

Author Response

I greatly appreciate your suggested pointers and corrections to the text.

In accordance with your comments and suggestions, I have made significant revisions to the manuscript.

Line 62: Why the fungus that lived in Antarctica produced stable lipases and biosurfactants is a great mystery.
This fungus seems to be essentially a phytopathogen, but why such a fungus was living in Antarctica is also a great mystery.

Lines 63-66: The results of Dr. Hoshino's and Dr. Oono's research can be found in the cited references.
Here I would like to highlight of their resumed fungal surveys in the Syowa Station area for the first time in about 50 years.

Line 98: Following your comment, I have corrected the relevant part to "narrow space".

Lines 162-164:In accordance with your comment, I have corrected the text in lines 156-162 in the new text.

Lines 192-193: I have refrained from using it in this manuscript because the Word formatting is corrupted when I create formulas in the Word options on my PC.

Round 2

Reviewer 2 Report

Some considerations were made previously, and nothing was changed in the text.

The title suggests a broad review about the diversity of cultivable fungi at the poles (Antarctic and Arctic), in addition to the survival and enzymatic production at sub-zero temperatures of these organisms, relating all of this with the impacts of climate change. However, the results presented and discussed refer to a limited number of collections and locations, carried out mainly by the author himself (44.4% of the cited references are self-citations). 

The introduction should be improved. For example, there is no reference in the first paragraph. The third paragraph is too short. Discuss more about the Arctic and put references.

Avoid using the personal pronoun, as in line 71 “we conducted” and line 235 "I would like".

Considering the results presented in the review and the amount of studies developed in the entire Antarctic continent that were not reported here by this review, it would be prudent to rewrite the sentence in line 38: "This review presents a history of fungal research in Antarctica".

Minor editing of English language is required.

Author Response

Thank you for your comments

  1. The research presented in this paper is clearly described in the Abstract and Introduction.
  2. I am the only person in the world who continues to conduct research on fungi at Syowa Station, Antarctica and in the Canadian High Arctic.
  3. Since no other reviewers have commented on the introduction, I consider the first round of reviews to be a sufficient improvement.
  4. Most sentences beginning with I were corrected to passive sentences.
  5. "This review presents a history of fungal research in Antarctica" was corrected to "This review presents the history of fungal surveys near the Syowa Station.

Reviewer 3 Report

The previous reviewer's recommendations were overlooked  and changes to the manuscript are still required:

1. The aims should be further stressed. The author promises to summarize the changes in fungal communities in glacial retreat areas and to point out the future directions of Antarctic and Arctic fungal research. None of these is made

2- Adaptations to formal writing (written in the third person, please consider rewriting sentences to remove the personal pronoun I) must be contemplated.

The previous reviewer's recommendations were overlooked  and changes to the manuscript are still required:

1. The aims should be further stressed. The author promises to summarize the changes in fungal communities in glacial retreat areas and to point out the future directions of Antarctic and Arctic fungal research. None of these is made

2- Adaptations to formal writing (written in the third person, please consider rewriting sentences to remove the personal pronoun I) must be contemplated.

Author Response

Thank you for your comments.

  1. "The author promises to summarize the changes in fungal communities in glacial retreat areas and to point out future directions of Antarctic and Arctic fungal research." is described in sections 4, 5, and 7.
  2. Most of the sentences that begin with "I" have been corrected to passive sentences.